# Diagnostic Methods in Forensic Pathology: Autoptic Findings and Immunohistochemical Study in Cases of Sudden Death Due to a Colloid Cyst of the Third Ventricle

**DOI:** 10.3390/diagnostics14010100

**Published:** 2024-01-01

**Authors:** Angelo Montana, Francesco Paolo Busardò, Giovanni Tossetta, Gaia Goteri, Pasqualina Castaldo, Giuseppe Basile, Giulia Bambagiotti

**Affiliations:** 1Department of Biomedical Sciences and Public Health, Università Politecnica delle Marche, 60126 Ancona, Italy; angelo.montana@ospedaliriuniti.marche.it (A.M.); g.goteri@univpm.it (G.G.); p.castaldo@univpm.it (P.C.); giulia.bambagiotti@gmail.com (G.B.); 2Department of Experimental and Clinical Medicine, Università Politecnica delle Marche, 60126 Ancona, Italy; g.tossetta@univpm.it; 3Trauma Unit and Emergency Department, IRCCS Galeazzi Orthopedics Institute, 20161 Milano, Italy; basiletraumaforense@gmail.com

**Keywords:** colloid cyst, sudden death, neurogenic stunned myocardium (NSM), immunohistochemical investigations, autopsy

## Abstract

The colloid cyst is a non-malignant tumor growth made of a gelatinous material covered by a membrane of epithelial tissue. It is usually located posterior to the foramen of Monro, in the anterior aspect of the third ventricle of the brain. Due to its location, it can cause obstructive hydrocephalus, increased intracranial pressure, and sudden cardiac death, catecholamine-mediated, through hypothalamus compression. All the mechanisms are still controversial, but the role of catecholamine has been confirmed with histological findings that highlighted myocardial injury (coagulative myocytolysis and contraction band necrosis, CBN). This study presents a case of sudden death in a previously healthy 22-year-old male due to a colloid cyst of the third ventricle. A complete autopsy was performed, highlighting in the brain an abundant quantity of cerebrospinal fluid (CSF) and a 2 cm pale grayish-green rounded cyst formation partially filling and distending the third ventricle. The diagnosis was confirmed through immunohistochemical investigation: positivity for Periodic acid-Schiff (PAS) staining and CK7 expression. In cases such as the one reported here, a combined approach of autopsy, histology, and immunohistochemistry is mandatory in order to identify the neoformation’s location and morpho-structural characteristics for a correct differential diagnosis, as well as to identify the cause of death.

## 1. Introduction

The colloid cyst is a benign epithelial-lined, fluid-filled cyst commonly located in the third ventricle or near the Monro’s foramen, which is at the anterior part of the third ventricle of the brain. The gelatinous material commonly contains mucin, old blood, cholesterol, and ions [1]. Colloid cysts are currently asymptomatic and identified as incidentaloma during imaging but sometimes can cause intermittency of symptoms like headaches, diplopia, memory issues, deficits, behavioral disturbances, and vertigo. Colloid cysts have rarely been cited as a cause of sudden death. This symptomatic spectrum and its intensity are dictated by cyst size and characteristics, ventricular size, and patient age. Colloid cysts become symptomatic when they enlarge rapidly and cause headaches, nausea, and vomiting secondary to obstructive hydrocephalus. Obstructive hydrocephalus occurs as a result of a blockage in the flow of cerebrospinal fluid (CSF) from the lateral ventricles to the third ventricle through the foramen of Monro [2]. A colloid cyst can act as a ball valve, physically blocking CSF outflow from the lateral ventricles. If this occurs, CSF accumulates into the lateral ventricles and causes ventriculomegaly and obstructive hydrocephalus. For this reason, some have hypothesized that colloid cysts may cause intermittent symptoms. However, the mechanism(s) of death is still controversial, underpinning several mechanisms that cause the sudden onset of severe symptoms followed by a rapid aggravation, occasionally fatal, of clinical conditions in patients with colloid cysts [3]. Several authors described the phenomenon of neurogenic stunned myocardium (NSM) as the cause of death. Here, we report a case of unexpected death in a previously healthy young man with an unknown colloid cyst of the third ventricle. We investigated the possible mechanism(s) of death through an accurate histological and immunohistochemical examination of the brain, heart, and cyst.

## 2. Case Report

A 22-year-old Somali man was found dead in the evening without visible signs of violence; therefore, the forensic pathologist was asked to examine the cause of his death. This study was in accordance with the 1964 Declaration of Helsinki or the institution’s ethical standards, subject to informed consent obtained from relatives.

### 2.1. Circumstantial Data

The young man, a guest in a reception facility for foreign refugees, was healthy and of normal weight. All available court files were analyzed. The general practitioner reported that he had normal blood pressure values, he was not taking any drugs, and he never suffered from headaches before.

On the morning of the day of his death, he suddenly developed a severe headache that forced him to go to bed in the early afternoon. His roommates monitored his condition throughout the afternoon and evening, and he still had headaches, nausea, and a single episode of unstoppable vomiting, then he rested again. At 10:00 p.m., he was found unconscious, so his roommates called an ambulance. Emergency physicians confirmed the death status and informed police authorities of the sudden and unexpected death. The local prosecutor instructed the forensic pathologist to investigate the crime scene and perform an autopsy to exclude any murder intention from others.

### 2.2. Autopsy

A complete autopsy was performed about 30 h after death. The body was refrigerated at 4 °C until autopsy. The external examination was unremarkable. The examination of the head and neck was performed through a posterior approach; this dissection allowed for cervical medulla, pons, cerebellum, and brain integrity. At the gross examination, the brain was symmetrical, enlarged, and edematous, weighing 1425 g. Marked brain swelling with flattening of the gyri and sulcal narrowing were also observed. The cerebellar tonsils were both grooved. The cerebral ventricles were slightly dilated and contained abundant amounts of clear and colorless cerebral spinal fluid (CSF) and an anomalous transparent membrane. The midbrain was free of hemorrhagic lesions. The brain was also fixed and examined after fixation according to the Ludwig technique. Internally, organs were in situs solitus, and no fluid was found in the pleural or abdominal cavities. During the autopsy, the larynx and trachea were opened and were filled with blood and mucus. The autopsy excluded the occurrence of acute and significant macroscopic abnormalities in all chest and abdomen organs.

The brain was collected in toto, and fragments of all organs after collection were preserved in formaldehyde and then stained with Hematoxylin Eosin for histological analysis.

### 2.3. Gross Examination of the Brain after Formalin Fixation

The successive gross examinations of the brain set for three weeks was conducted by sectioning the specimen in the horizontal plane. There was a 2 cm pale grayish-green rounded cyst formation partially filling and enlarging the third ventricle (Figure 1A), which was wedged between the fornix splayed columns, non-occluding the left foramen of Monro (Figure 1B). The lateral ventricles were slightly distended. Sectioning the cyst formation revealed a viscous substance indurated after formalin fixation with a thin fibrous capsule (Figure 1C,D).

### 2.4. Histological and Immunohistochemical Analysis of Colloid Cyst

Periodic acid-Schiff (PAS) staining (specifically for the identification of mucin) of the cyst formation showed a cystic lesion filled with a strongly PAS-positive amorphous content and covered by a hypocellular fibrous wall and by a simple columnar epithelium with a variable amount of cilia or mucin, flattened without squamous metaplasia (Figure 2A,B). A histological examination of the brain revealed a severe edema. The other organs showed signs of central dysregulation (pulmonary edema).

In order to verify whether it was a cyst or cancerous tissue, we performed an immunohistochemical analysis and found that the specimen was positive for cytokeratin 7 (CK7) expression (Figure 3), a structural protein expressed in epithelia lining the cavities of internal organs, while it was negative for ALCIAN, CK20, CEA, Cam 5.2, TTF1, CD68, and S-100 expression. These findings lead to the diagnosis of a colloid cyst.

### 2.5. Histological Analysis of Myocardial and Brain Tissue

Paraffin-embedded blocks were made after processing, and sections were cut approximately 2–3 microns thick and stained routinely using hematoxylin-eosin and Perls’ stain. Myocardial tissue blocks showed myocell fragmentation and pathological band with hyper eosinophilia of the hypercontracted myocardial cells and myofibrils rhexis into cross-fiber (Figure 4). Edema, hemorrhage, and myofiber vacuolization were absent. All brain blocks showed signs of hypoxia with diffuse pericellular and perivascular edema and enlarged neurons.

### 2.6. Toxicological Analysis

Peripheral blood and urine samples were collected during the autopsy, stored in a −20 °C freezer, and then processed for toxicological analysis. After the extraction, the samples (100µL of whole blood and 100µL of urine) were analyzed using a high-performance liquid chromatography–tandem mass spectrometry (HPLC–MS/MS) screening method. Through this analytical technique, a broad spectrum of compounds can be identified, including new psychoactive substances, classic drugs, and their metabolites. In detail, 100 µL of whole blood spiked with a mixture of internal standard deuterated was added to 70 µL of M3^®^ reagent, and after precipitation, the supernatant was evaporated to dryness and reconstituted with 1 mL of mobile phase. Then, 100 µL of urine spiked with internal standard deuterated mix was diluted with 500 µL of M3^®^ buffer [4]. Finally, one microliter of each sample was injected into HPLC–MS/MS for the analysis. No substances were detected in the analyzed blood and urine samples.

## 3. Discussion

Through macroscopic and histological findings of the cyst and heart, we can state that death was due to the presence of a colloid cyst of the third ventricle. Dissection of the brain revealed a 2 cm cystic lesion inside the third ventricle. The lateral ventricles were mildly dilated. Cyst sections revealed a viscous substance hardened after formalin fixation with a thin fibrous capsule. In this context, the contribution of immunohistochemistry was fundamental. Immunohistochemical stains were performed on the greenish neoformation in order to differentiate between colloid cysts and cancer. These stains, specifically for mucin, cytokeratin, and tumor markers, were PAS, ALCIAN, S100, CK7, CK20, CEA, Cam 5.2, TTF1, and CD68. It was positive for PAS staining and CK7 protein, a cytoplasmic cytokeratin.

Colloid cyst is a congenital endodermal malformation containing mucous and gelatinous material usually located in the third ventricle; however, other locations may include leptomeninges, the frontal lobe, the cerebellum, brain convexity, the brainstem, the fourth ventricle, and the region of the optic chiasma [5,6,7,8,9,10,11,12]. Clinically, colloid cysts are usually discovered incidentally in childhood and adolescence and remain asymptomatic until the third or fourth decade of life; however, symptoms can appear at any age. They are generally round in shape, ranging from a few millimeters to several centimeters in size, and are histologically benign. Normally, symptomatic cysts have a diameter of 1–2 cm, and clinical symptoms attributable to increased intracranial pressure are not specific.

A scientific literature research was performed from 1990 to 2023, using the following terms: “sudden death”, “colloid cyst”, and “immunohistochemistry”, and a combination thereof. In the literature, there were ten cases of death due to a colloid cyst and eight non-fatal cases with healthy patients after neurosurgery. In Table 1, a summary of the details of the included research articles is reported.

Although obstructive hydrocephalus is a common postmortem finding, not every patient with a colloid cyst showed ventricle dilation [20].

Jarquin-Valdiviaa AA et al. [15] reported the case of a 33-year-old woman with an intraventricular cerebral colloid cyst who survived after a cardiac arrest due to an acute hydrocephalus. Sudden death in patients with colloid cysts may be related to an acute neurogenic cardiac dysfunction induced by a vigorous activation of the cerebroprotective neuroendocrine system, in turn, due to a sudden decrease in cerebral perfusion pressure caused by an increased intracranial pressure.

Demirci S. et al. [16] reported three cases of sudden death due to a colloid cyst of the third ventricle. In all three cases, during the autopsy, they observed a grey transillumination area in the base of the brain, between the chiasma opticum and the corpus mamillare. They highlighted the importance of brain dissection and examination during the autopsy in cases where the clinical history is characterized by headaches and the difficulty in the differential diagnosis in young adults.

Büttner et al. [5] reported two cases of a fatal third ventricle colloid cyst, and reviewing the literature, they found 98 cases (40 females, 41 males, 17 sex not given) of sudden death due to a colloid cyst of the third ventricle. The mean age of patients was 29.6 years, ranging from 6 to 79 years; the cyst size ranged from 0.8 cm × 0.8 cm, the smallest, to 7.9 cm × 6.5 cm, the largest; the onset of symptoms ranged from 17 years to a few hours prior to death.

E. Turillazzi et al. [17] described one 10-year-old boy’s death after the sudden onset of severe headache and vomiting attacks. He lost consciousness after being visited by an emergency physician, and he was taken to the local hospital, where he was dead on arrival. A 2 cm light-pale grayish-white cyst that was partially filling and distending the third ventricle and located between the fornix’s splayed columns was discovered during the autopsy. It did not obstruct the left foramen of Monro, but the lateral ventricles were mildly distended. When the cyst was sectioned, a viscid substance was surrounded by a thin fibrous capsule and hardened after formalin fixation. They attributed the death to acute cardiac arrest due to hypothalamus stimulation. Specifically, very close to the walls of the third ventricle, which is the most frequent anatomical site of colloid cysts, are hypothalamic structures, which seem to be involved in neuroendocrine and autonomic regulation and in cardiovascular control. Compression of the hypothalamic cardiovascular regulatory centers by the cyst leads to reflex cardiac effects, which may explain the mechanism of death in cases of sudden death in patients with a colloid cyst without signs of hydrocephalus or brain herniation.

Torrey J. et al. [13] described the case of an 11-year-old boy who, 18 days after a dance event where he repeatedly shook his head back and forth (“head-banged”), died. In detail, after approximately 5 min of “head-banging”, he suddenly felt sick, and from two days later until his death, he had a severe headache. The autopsy revealed cyst apoplexy associated with cyst expansion and ventriculomegaly.

Cuoco J.A. et al. [19] reported a case of sudden coma and death due to a hemorrhagic third ventricular colloid cyst in a 21-year-old male who was jogging. The patient had an acute onset of severe headache associated with nausea and vomiting that, in a few hours, developed into a coma. Imaging showed a hemorrhagic colloid cyst of the third ventricle, so the authors hypothesized that the rise in systolic blood pressure due to exercise caused the colloid cyst’s vessels to rupture, resulting in hemorrhage. The patient suffered from cyst apoplexy, which caused an increase in cyst dimensions and the occlusion of the foramina of Monro, resulting in an acute onset of obstructive hydrocephalus and catastrophic neurologic sequelae.

Nelson E. et al. [21] also described three cases of a fatal colloid cyst associated with air travel. In particular, two patients developed symptoms of acute hydrocephalus during the flight, while the third case involved a pilot who died suddenly immediately after landing.

Sudden death occurred in about 10% of patients with a third ventricle colloid cyst. Acute deterioration occurs in 34% of cases, with a mortality rate of 12% in the case of a symptomatic colloid cyst [18]. Fatal intracranial hypertension, obstructive hydrocephalus, and acute lateral ventricle dilatation are due to intermittent or persistent obstruction of Monro’s foramen; however, the compression of the hypothalamus by the cyst can also lead to death, causing a cardiac reflex [22]. In this case, as regards the death mechanism, a dual combined mechanism can be hypothesized. It is known that hypothalamic structures are located in proximity to the walls of the third ventricle and hypothalamic nuclei (e.g., arcuate nucleus) and are involved in cardiovascular regulation [23]. Consequently, it is possible to hypothesize that protracted hypothalamic pressure exerted by the adjacent colloid cyst on the hypothalamus may have caused abnormal autonomic signaling resulting in the release of catecholamine, a phenomenon called neurogenic stunned myocardium (NSM) [15]. NSM has been described in many neurological disorders, such as subarachnoid hemorrhage (SAH), hemorrhagic and ischemic stroke, multiple sclerosis, head injury, brain tumors, meningitis, encephalitis, spinal cord lesions, hydrocephalus, psychological stress, and intracranial hypertension [24,25]. An increase in cerebral perfusion pressure is a normal physiologic response to the development of acutely elevated intracranial pressure. Sympathoadrenal hyperactivity results in acute, severe arterial hypertension in the setting of elevated ICP.

Ryder JW et al. [26] proposed that the direct stimulation of cardiac reflex centers may play a role in sudden deterioration and death. The location of the third ventricle colloid cysts and the resulting increase in intracranial pressure appear to stimulate the hypothalamic regulatory centers involved in neuromediated cardiovascular control. There is strong evidence that the cardiac changes observed in neurological catastrophes are linked to the overactivity of the sympathetic limb of the autonomic nervous system being overactive. These cardiac effects may contribute to the mortality rates of many other neurological conditions, primarily cerebral infarction, subarachnoid hemorrhage, status epilepticus, and head trauma. These phenomena may also be important in the pathogenesis of sudden death due to catecholamine toxicity following compression of the hypothalamus by the colloid cyst.

The development of cardiac anomalies after neurological events, such as seizures, stroke, etc., is known as NSM. These occurrences lead to autonomic nervous system dysfunction, which in turn can induce various types of cardiac disorders [27,28]. The increase in catecholamine levels due to brain autonomic dysfunction is the mechanism underlying NSM. In NSM, despite the similar presentation with myocardial infarction, there is not any significant coronary artery blockage. In NSM, as histological changes, there is myocytolysis, which is transient myofibrillar degeneration characterized by multiple foci of subendocardial hemorrhage around epicardial nerves and indicative of myocardial stress. In NSM, arrhythmias are frequently observed, as myocytolysis affects the subendocardial region, which is also the site of the cardiac conduction system. The initial event in the NSM is a catecholamine influx due to either excessive stress or a brain injury. Cardiac damage occurs in three ways: (a) ischemic insult due to the inability to handle the increased myocardial demand [29]; (b) direct myocardial toxicity due to increased catecholamine levels; and/or (c) coronary vasospasm as a result of elevated catecholamine levels. Direct toxicity is due to an increase in calcium influx in the cardiac muscle cells. This increase occurs because beta-adrenergic receptors, stimulated by increased catecholamine levels, lead to the opening of calcium channels. Increased calcium influx into cardiomyocytes causes contractile dysfunction and reduces adenosine triphosphate (ATP) levels, eventually leading to cell death. The catecholamines can induce myocardial damage through (1) free radical-mediated lipid peroxidation with intramyocellular Ca^2+^ influx and production of reactive oxygen species (ROS) that played in the pathophysiology of CBN; (2) interaction with adrenergic receptors [30,31]; (3) effect of ROS on the expression of TNF-α (tumor necrosis factor-alpha), MCP-1 (monocyte chemotactic protein-1), interleukins IL6, IL8, and IL10, and a significant randomly scattered apoptotic process in the damaged myocardium [32,33]. In the present case, compression of the hypothalamus by the third ventricle colloid cyst may have led to sudden death through two mechanisms. The first proposed cascade consists of catecholamines triggering calcium channel activation, release of reactive oxygen species, and peroxidation of lipid membranes, all of which lead to contraction band necrosis and death. The second suggested cascade involves a catecholamine spike causing reflex cardiac effects, including increased cardiac preload and afterload; the increase in ventricular inotropy, dromotropy, and chromotropy; and increased cardiac irritability leading to arrhythmias. These reflex cardiac effects may lead to neurogenic myocardial stunning and sudden death.

Along with the above mechanism, death may also be due to a sudden onset of obstructive hydrocephalus and a massive increase in intracranial pressure. The colloid cyst may, in fact, cause a physical obstruction to the normal flow of cerebrospinal fluid that normally circulates from the lateral ventricle to the third ventricle through the foramen of Monro [34]. According to Monro–Kellie’s law, the skull is a closed box containing CSF, brain, and blood vessels. The balance among these is critical for maintaining proper endocranial pressure and brain physiology. According to this law, if there is a change in one of these three components, the others necessarily must adapt by reduction. Physical blockage to CSF outflow leads to accumulation of CSF at the intracranial level with increased intracranial pressure. In the case of a significant increase in intracranial pressure, there is dislocation of the brain with cerebral hernia formation (transtentorial, subfalcine, or tonsillar hernia) and cessation of cerebral perfusion caused by compression of vessels. Hypoperfusion of brain tissue causes brain atrophy, permanent brain damage, and possible death, while the formation of brain hernias results in brainstem compression. In particular, transtentorial (uncal) herniation compresses the midbrain and tonsillar herniation of the bulb, causing sudden death due to cardiorespiratory arrest [17,35,36].

When evaluating cases of sudden death due to an undiagnosed colloid cyst, there were several problems in defining the most probable causes of death. Although modern diagnostic imaging techniques have revolutionized the diagnosis of brain tumors, autopsy, careful gross examination, and fixed brain section (with coronal section) are still essential in determining the exact location, topography, mass effects, the histological characteristics of the neoformation, secondary brain damage and especially the cause of death [37]. In these cases, it is also necessary to perform immunohistochemical analysis for a correct differential diagnosis.

In the end, considering the autopsy and the histological and immunohistochemical findings, we can state that the cause of death was due to a combined double mechanism resulting from the presence of a third ventricular colloid cyst: an obstructive hydrocephalus with increased intracranial pressure, originating from the physical encumbrance of the colloid cyst, and the reflex cardiac effects mediated through the compression of the hypothalamus by the cysts (Figure 5).

## Figures and Tables

**Figure 1 diagnostics-14-00100-f001:**
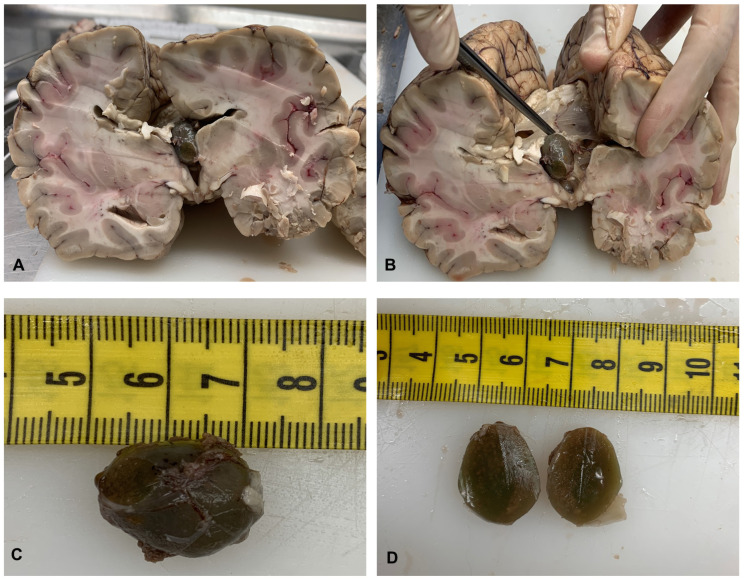
Localization of the cyst in the third ventricle (**A**). The cyst was wedged between the splayed columns of the fornix, not obstructing the left foramen of Monro (**B**). A viscous substance hardened after formalin fixation with a thin fibrous capsule was observed when the cyst was sectioned (**C**,**D**).

**Figure 2 diagnostics-14-00100-f002:**
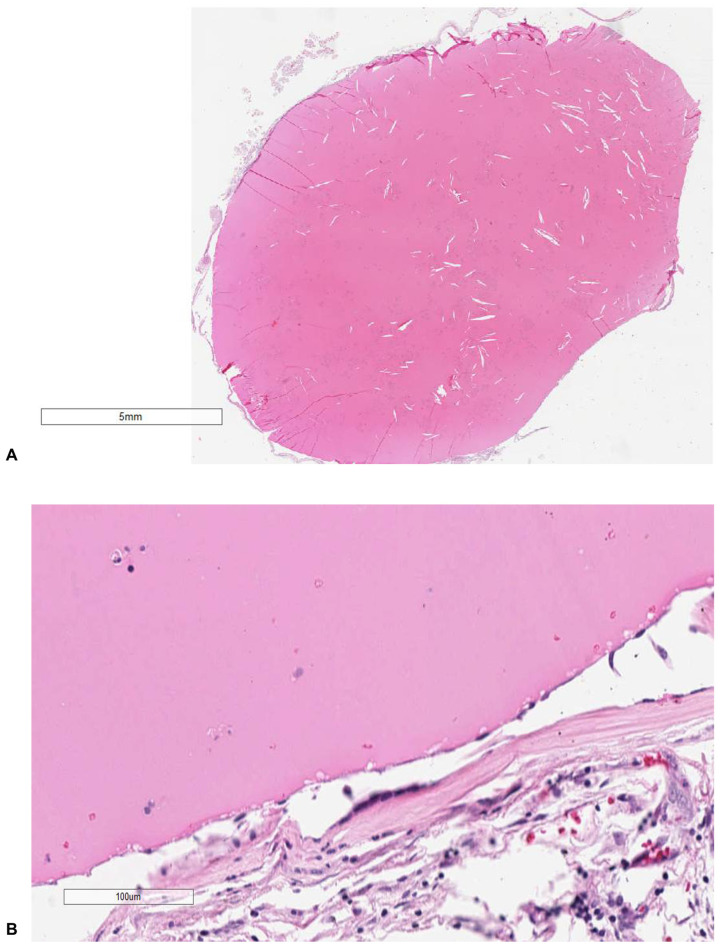
Cystic lesion, filled with a strongly PAS-positive amorphous content (**A**) and covered by hypocellular fibrous wall lined by simple columnar epithelium with variable cilia or mucin, flattened without squamous metaplasia (**B**). Columnar, ciliated epithelial cells compose the cyst wall. There was no abnormal increase in goblet cells or any cellular atypia or thickening of the wall. Cyst content consisted of a mucoid material with detached epithelial cell clusters (**A** ×40, **B** ×80).

**Figure 3 diagnostics-14-00100-f003:**
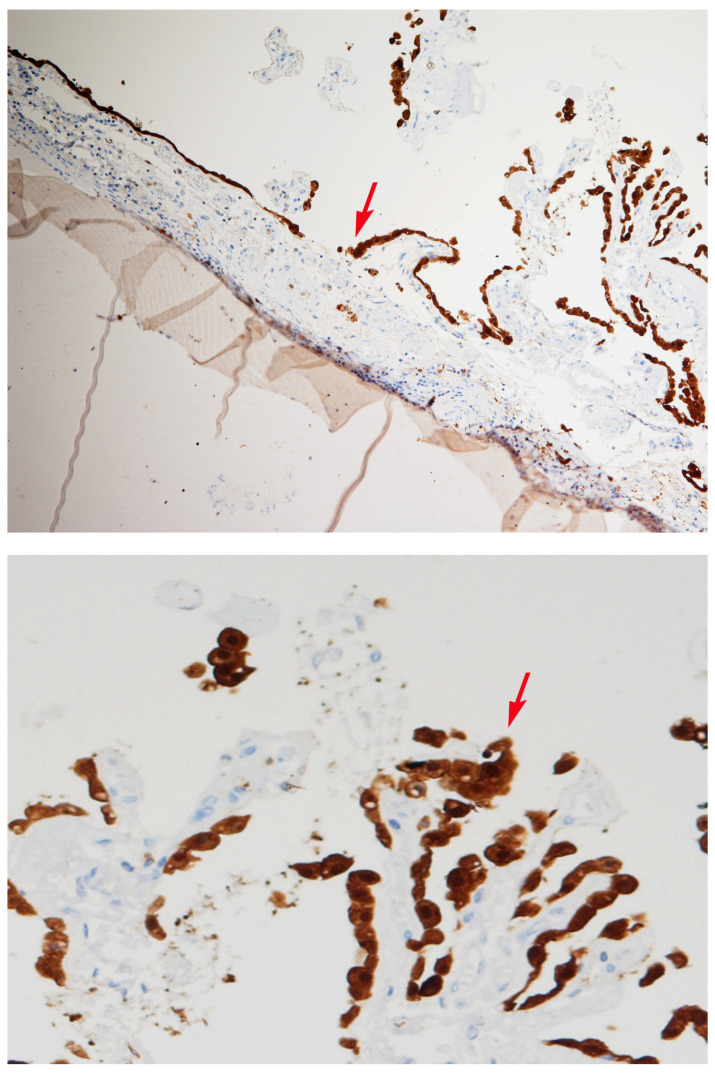
Immunohistochemical staining of CK7 using a human-specific anti-Cytokeratin 7 antibody. The cyst epithelium (arrows) is positive for CK7 (×40, ×80).

**Figure 4 diagnostics-14-00100-f004:**
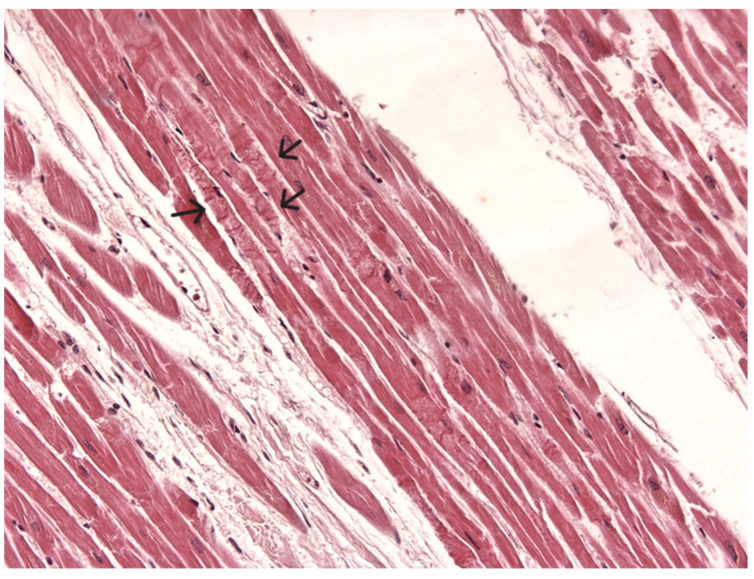
Hypercontracted myofibrils fragmentation and hypercontracted or coagulated sarcomeres (arrows) band formation. Absence of edema, hemorrhage, and myofiber vacuolization (H&E × 40).

**Figure 5 diagnostics-14-00100-f005:**
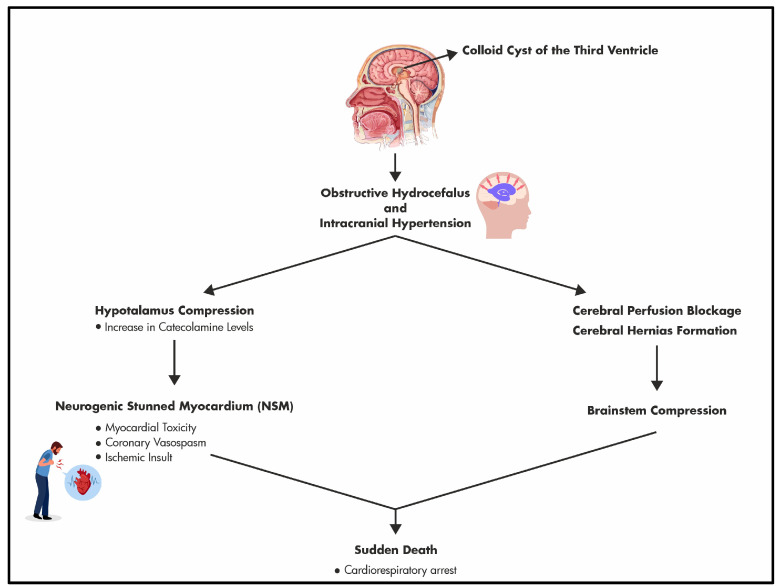
Combined double mechanisms underlying the sudden death due to a colloid cyst of the third ventricle.

**Table 1 diagnostics-14-00100-t001:** Sudden death and non-fatal cases related to colloid cysts of the third ventricle.

Age (Years) Sex	Cyst Size (cm), Localization	Symptoms and Onset	Cause of Death	Histology Findings/Immunohistochemistry	Authors
11 M	1.5 cm in diameter; inside the third ventricle	18-day history of severe headache after 5 min of “head-banging”		Cyst: blood clot with central hemolysis, encapsulated by fibrin and “debris” containing hemosiderin-laden macrophages/ Strongly PAS positive	Torrey (1983) [13]
21 F	In the fourth ventricle and extended to the inferior part of the third ventricle	3-year history of intracranial hypertension, occipital headache radiating to the cervical region, nausea, occasional vomiting	Underwent neurosurgery		Jan et al. (1989) [12]
33 F	In the right frontoparietal cerebral hemisphere	Focal epilepsy involving the left arm	Underwent neurosurgery	Cyst wall: cuboidal and ciliated columnar epithelial lining	Campbell et al. (1991) [8]
40 F	1 cm in diameter, inside the third ventricle near the foramen of Monro	Vomit and frontal headache after an intercontinental air flight	Brainstem compression caused by obstruction of the third ventricle	Cyst: lined by a single layer of flattened, non-ciliated epithelial cells, resting on a collagenous membrane/Strongly PAS-positive Brain: edema, hypoxic nerve cell changes, moderate nerve cell loss	Büttner et al. (1997) [5]
33 M	4 × 3.5 × 3.5 cm, inside the third ventricle	Recurrent headaches		Cyst: lined by a single layer of partially low cuboidal, partially flattened, non-ciliated, and ciliated epithelial cells, resting on a thin collagenous membrane/Strongly PAS positive Brain: extensive edema, hypoxic nerve cell changes	
45 F	5 × 4 cm, extending dorsally to the surface of the cerebellum, medially to the cerebellar vermis, pressing the fourth ventricle	Signs of increased intracranial pressure, diffuse headache, gait disturbance and nausea	Underwent neurosurgery	Cyst: single layer of partly ciliated columnar epithelial cells resting on a basal lamina/Strongly PAS, collagen type IV, EMA, and S-100 protein positive	Müller et al. (1999) [9]
34 M	5 cm in diameter, in the parietal region	Seizures	Underwent neurosurgery	Cyst: stratified ciliated columnar epithelium	Efkan et al. (2000) [10]
15 F	At the ventral portion of the ponto-mesencephalic junction	3-month history of headache and diplopia	Underwent neurosurgery	Cyst: collagenous capsule lined by a single layer of non-ciliated epithelium that ranged morphologically from cuboidal to slightly columnar/Positive immunoreactivity to cytokeratin	Inci et al. (2001) [11]
36 F	Optic chiasm	Sensorimotor hemi-syndrome, retrobulbar neuritis, acute asymmetric upper nasal quadrantanopsia	Underwent neurosurgery	Cyst: columnar and ciliated epithelial lining	Killer et al. (2001) [14]
33 F	1.2 cm, at the foramen of Monro in the third ventricle	3- to 5-day history of headache, neck pain, nausea, emesis	Underwent neurosurgery		Jarquin-Valdivia et al. (2005) [15]
17 F	1 cm, inside the third ventricle	2-year history of intermittent headaches	Dilated ventricular system secondary to blockage of the third ventricle, intrapulmonary hemorrhage associated with congestion and edema		Shaktawat et al. (2006) [7]
60 M	5 cm in diameter, in the left frontal lobe	Asymptomatic	Underwent neurosurgery	Cyst: inside the third ventricle/ Positive for cytokeratin end epithelial membrane antigen (EMA)	Tanei et al. (2006) [6]
24 F	2 cm in diameter, inside the third ventricle near the foramen of Monro	12-month history of migraine headaches	Sudden death due to colloid cysts of the third ventricle	Cyst: a single layer of non-ciliated columnar epithelial cells with pale eosinophilic cytoplasm resting on a fibro-collagenous membrane/ Strongly positive for mucicarmine staining	Demirci et al. (2009) [16]
21 F	2 cm in diameter, in the third ventricle	6-month history of migraine headaches			
25 M	2.5 × 2.5 × 2 cm, inside the third ventricle	Headaches and vomiting for 3 days before the death			
10 M	2 cm in diameter, filling the third ventricle	Severe headache, vomiting attacks	Acute cardiac arrest due to hypothalamus stimulation	Cyst: single stratum of cuboidal, semi-flattened, unciliated, and ciliated epithelial cells lying on a thin collagenous membrane/ Strongly PAS positive Brain: severe edema Heart: hypercontracted myocardial cells, short sarcomeres, very thick Z lines, myofibrils rhexis into cross-fiber	Turillazzi et al. (2012) [17]
22 F	1.22 × 1.47 × 1. 23 cm, at the foramen of Monro	2-days of global bitemporal headache, nausea and vomiting	Brain dead 7 days after		Al-Hashel et al. (2015) [18]
21 M	1.3 cm in diameter, in the third ventricle	Acute onset of a severe headache, nausea, vomiting, and past migraines in the last months	Brain dead after 5 days for acute obstructive hydrocephalus, cerebral edema, and uncal herniation.		Cuoco et al. (2018) [19]

## Data Availability

The data presented in this study are available on request from the corresponding author. The data are not publicly available due to privacy.

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
