# Peer review of "Diagnostic Methods in Forensic Pathology: Autoptic Findings and Immunohistochemical Study in Cases of Sudden Death Due to a Colloid Cyst of the Third Ventricle"

_diagnostics, 2024, doi:10.3390/diagnostics14010100_

Round 1

Reviewer 1 Report

Comments and Suggestions for Authors

Dear Authors 

The presented article concerns a scarce kind of death related to brain colloid cyst complications revealed in a post-mortem examination. Since the literature in this area is sparse and the issues are fascinating, I propose publishing the article with some changes.

The colloid cyst of the third ventricle can be fatal because of acute cardiac arrest due to hypothalamus stimulation. The pathological cardiac event can also respond to increased intracranial cerebral pressure in the context of the third ventricle colloid cyst. The cyst can obstruct the flow of cerebrospinal fluid and lead to prominent hydrocephalus. Acute ventricular hydrocephalus with intracranial hypertension can result in brain stem compression and death. 

I found an article (Turillazzi et al. Diagnostic Pathology 2012, 7:144 http://www.diagnosticpathology.org/content/7/1/144) where possible death mechanisms were clearly presented. I suggest discussing possible mechanisms of death in more detail, not only in two sentences (a graphic diagram is also welcome). 

Finally, the section "Peripheral blood and urine samples were collected, stored at -20 °C, and then pro-96 ceased for toxicological analysis by applying a previously published HPLC- 97 MS/MS method" looks unfinished. The results of the analyses should also be visible in the article. Therefore, I propose another section on toxicological test results: 2.7. Toxicological analysis.  

Sincerely 

Reviewer 

Author Response

Reviewer #1: The presented article concerns a scarce kind of death related to brain colloid cyst complications revealed in a post-mortem examination. Since the literature in this area is sparse and the issues are fascinating, I propose publishing the article with some changes.

  1. The colloid cyst of the third ventricle can be fatal because of acute cardiac arrest due to hypothalamus stimulation. The pathological cardiac event can also respond to increased intracranial cerebral pressure in the context of the third ventricle colloid cyst. The cyst can obstruct the flow of cerebrospinal fluid and lead to prominent hydrocephalus. Acute ventricular hydrocephalus with intracranial hypertension can result in brain stem compression and death. I found an article (Turillazzi et al. Diagnostic Pathology 2012, 7:144 http://www.diagnosticpathology.org/content/7/1/144) where possible death mechanisms were clearly presented. I suggest discussing possible mechanisms of death in more detail, not only in two sentences (a graphic diagram is also welcome).

Response: We thank the reviewer for the suggestion. We have explained the dual mechanism underlying death more in detail in the discussion section, considering the reviewer's proposed article (already reported in the references section - n. 17). We have also included a graphic diagram.

  1. Finally, the section "Peripheral blood and urine samples were collected, stored at -20 °C, and then pro-96 ceased for toxicological analysis by applying a previously published HPLC- 97 MS/MS method" looks unfinished. The results of the analyses should also be visible in the article. Therefore, I propose another section on toxicological test results: 2.7. Toxicological analysis.

Response: According with the reviewer’s comment we add the section “2.7. Toxicological analysis”.

Reviewer 2 Report

Comments and Suggestions for Authors

The article describes a rare and interesting case of sudden death caused by a colloid cyst. The article is interesting, mostly well structured and written. The authors also review the relevant literature.

There is one structural error: there are two No. 3. sections – „Cause of Death” and „Discussion”. It needs to be corrected.

I think it would be good to know when the symptoms started (the article states only in line 68 that „during the day of his death”).

There is only one important aspect that has to be clarified. The authors are focusing on NSM as a mechanism of death but do not explain clearly why they rule out the possibility of a herniation and respiratory depression (even it is not clear whether they rule it out). I think – based on the autopsy findings (severe brain edema, grooved cerebellar tonsills) and clinical course (headache, nausea and „unstoppable” vomiting and possible somnolence at the end), it would be a well-established mechanism of death. In lines 153-156, the authors also mention a possible double mechanism - I absolutely agree with this possibility.

My proposition is that the subsection „cause of death” (lines 151-156) should be moved to the end of the discussion as a summary. The possible role of the intracranial pressure increase and herniation should also be discussed in the discussion section.

I recommend the publication of the article after a minor revision.

Comments on the Quality of English Language

I have only a few suggestions about language quality:
line 53: as the cause of death
line 54: with an unknown
line 72: the death status
line 90: were opened
line 105: examination of the brain fixed for three weeks

Author Response

Reviewer #2: The article describes a rare and interesting case of sudden death caused by a colloid cyst. The article is interesting, mostly well structured and written. The authors also review the relevant literature … I recommend the publication of the article after a minor revision.

  1. There is one structural error: there are two No. 3. sections – „Cause of Death” and „Discussion”. It needs to be corrected.

Response: Amended. We move the section “Cause of death” to the end of discussion as a summary.

  1. I think it would be good to know when the symptoms started (the article states only in line 68 that „during the day of his death”).

Response: Amended, according to the reviewer comment we modified the text.

  1. There is only one important aspect that has to be clarified. The authors are focusing on NSM as a mechanism of death but do not explain clearly why they rule out the possibility of a herniation and respiratory depression (even it is not clear whether they rule it out). I think – based on the autopsy findings (severe brain edema, grooved cerebellar tonsills) and clinical course (headache, nausea and „unstoppable” vomiting and possible somnolence at the end), it would be a well-established mechanism of death. In lines 153-156, the authors also mention a possible double mechanism - I absolutely agree with this possibility. My proposition is that the subsection „cause of death” (lines 151-156) should be moved to the end of the discussion as a summary. The possible role of the intracranial pressure increase and herniation should also be discussed in the discussion section.

Response: We thank the reviewer for the comment, we moved the section “Cause of death” to the end of discussion as a summary (lines 388-393) and we have explained the role of the intracranial hypertension and herniation more in detail in the discussion section. (lines 357-372)

  1. I have only a few suggestions about language quality:

line 53: as the cause of death

line 54: with an unknown

line 72: the death status

line 90: were opened

line 105: examination of the brain fixed for three weeks

Response: Amended, according to the reviewer comment we modified the text.